# Cancer Prevention: Knowledge, Attitudes and Lifestyle Cancer-Related Behaviors among Adolescents in Italy

**DOI:** 10.3390/ijerph17228294

**Published:** 2020-11-10

**Authors:** Gabriella Di Giuseppe, Concetta P. Pelullo, Maria Mitidieri, Giuseppe Lioi, Maria Pavia

**Affiliations:** Department of Experimental Medicine, University of Campania “Luigi Vanvitelli”, 81100 Naples, Italy; gabriella.digiuseppe@unicampania.it (G.D.G.); concettapaola.pelullo@unicampania.it (C.P.P.); maria.mitidieri@unicampania.it (M.M.); lioig84@gmail.com (G.L.)

**Keywords:** attitudes, cancer, knowledge, lifestyle, prevention, risk factors, lifestyle cancer-related behaviors, adolescents

## Abstract

This study explores knowledge, attitudes and lifestyle behaviors related to cancer in a sample of adolescents. Data were collected through a self-administered questionnaire. 871 adolescents agreed to participate, with a response rate of 96.8%. Only 26.1% had a good level of knowledge on most risk factors for cancer. Adolescents with both parents employed, with a personal, familiar or friend history of cancer or having received information about cancer prevention from a physician, were more likely to have good knowledge about the risk factors for cancer. In total, 41% of participants declared that they consumed alcohol and 25.3% declared they were current smokers, 19.2% consumed fruits or vegetables more than once a day and 75.2% reported poor physical activity. Older adolescents, with a personal, familiar or friend history of cancer, not having one parent in the healthcare sector or not physically active were significantly more likely to be current smokers, whereas physical activity was significantly more likely in adolescents who had been informed by physicians on cancer prevention, and had one parent in the healthcare sector. This study highlights a need for improved education of adolescents about cancer prevention and lifestyle cancer-related behaviors.

## 1. Introduction

Despite the improvement in the survival of many patients, cancer remains the main cause of mortality in many countries of the world and it is associated with an increasing global incidence [1], with serious repercussions on health expenditure, because of the resources used in pharmacological treatments, advanced therapies and palliative care [2]. In Italy, in 2019, approximately 371,000 new cases were estimated and there were more than a thousand new diagnoses of cancer per day [3].

The World Health Organization (WHO) has estimated that about one third of the diagnoses of cancer are attributable to lifestyle cancer-related behaviors, such as smoking habits, poor physical activity and nutrition, alcohol abuse and exposure to the sun without adequate protection [4]. Therefore, although the modifiable causes of cancer are various and complex, prevention remains the most cost-effective strategy to fight this disease [5]. 

Adolescence represents a sensitive period in which lifestyle cancer-related behaviors, such as smoking habits, alcohol abuse, poor physical activity and incorrect eating habits, are often acquired [6,7] and can continue into adulthood with a consequent increase of morbidity and mortality [8,9]. Therefore, providing information to adolescents about lifestyle cancer-related behaviors, driving them to more correct lifestyle choices, may be useful for developing effective prevention programs for a healthy adulthood [10]. 

In the literature, several studies investigating knowledge, attitudes and lifestyle cancer-related behaviors have been conducted in the adult population [11,12,13,14], while evidence on adolescents is more limited [10,15]. The aims of this survey were to explore knowledge and attitudes related to cancer risk factors among secondary school adolescents in Southern Italy, to acquire information regarding their lifestyle behaviors and to identify predictors associated with knowledge and lifestyle cancer-related behaviors.

## 2. Materials and Methods

### 2.1. Study Design

This survey was conducted from September 2019 to January 2020 among a random sample of 13 to 20 years old adolescents attending public secondary schools in the metropolitan area of Naples, Southern Italy. The sample was selected through a two-stage cluster sampling method. First, 7 schools were randomly selected from the list provided by the local education office including all eligible schools in the study area. In the second stage, a simple random sampling technique was adopted to select adolescents from each school. Sample size was calculated before study initiation, considering a prevalence of 50% of adolescents that knew recommendations about cancer prevention, a desired precision of 0.05 and a design effect of 2 [16]. According to this calculation a sample size of 768 subjects was needed.

### 2.2. Data Collection

Before starting the investigation, the deans of the selected schools were invited to participate in the survey through a letter, containing the objectives and the methodology of the study. After deans’ approval, parents of selected children received a letter asking the written permission for their children to participate in the study; the letter reported also the study goals, highlighted why their children were selected and assured them that confidentiality and anonymity of responses would be strictly guaranteed. Moreover, parents were also informed that the data would be analyzed anonymously, and that no payment would be received for participation in the study.

### 2.3. Questionnaire

The survey instrument was an anonymous and self-administered questionnaire. A member of the research team in each classroom gave instructions about the questionnaire and distributed it to the selected adolescents who had received parental consent. The questionnaire was divided into five sections. The first gathered information about adolescents’ socio-demographics characteristics, including age, gender and parents’ education level and working activity. Moreover, it was asked whether they had a personal, familiar or friend history of cancer. In the second section knowledge on cancer definition, risk factors, epidemiology as related to frequency, age of occurrence and association with smoking and alcohol consumption was explored. The third section examined the adolescents’ attitudes about cancer. Their beliefs about perceived severity and opportunity for prevention of cancer were measured on a 3-point Likert scale with options for agree, uncertain and disagree. Moreover, the perception of risk of developing cancer was measured on a 10-point Likert scale from 1 to 10, with 1 being “not at all worried” and 10 being “extremely worried”. The self-reported health status was also measured on a 5-point Likert scale ranging from 1 (very unsatisfactory) to 5 (very satisfactory), and body mass index (BMI) was calculated. 

The fourth section explored participants’ lifestyle cancer-related behaviors, according to the European Code against Cancer [17]. In particular, smoking status, alcohol consumption (Audit-C scale) [18], sedentary behavior (time spent watching television and using a computer) [19], physical activity (in minutes in a typical week) [20], dietary habits (fruit, vegetables and sugary drinks consumption) [21] and sun and tanning lamp exposure were investigated. The fifth section dealt with adolescents’ sources of information about cancer prevention, and whether they needed to receive additional information. Response options included “yes” or “no” answers. Items of the questionnaire are reported in the Appendix A.

BMI was calculated by dividing weight (in kg) by the square of the height (in m), using the percentile tables for the Southern Italian population [22] according to sex and age. Adolescents were categorized as “overweight” if BMI was between the 85th and the 94th percentile and “obese” if BMI was ≥95th percentile.

The instrument used to assess physical activity, sedentary and dietary behaviors was the Global School-based student health survey (GSHS) for this specific age group [19,20,21]. In particular, for physical activity we asked the time spent on this activity in the previous week, sedentary behaviors were calculated considering the total time spent on watching television and using a computer, and dietary habits evaluating the daily intake of fruit and vegetable consumption. Cut-off values to define these behaviors were derived by the WHO recommendations for adolescents [21]. Subjects were classified as physically inactive if they performed <300 min/week of physical activity, and physically active for ≥300 min/week, as having a sedentary behavior if they declared to spend ≥2 h per day watching television and using a computer and to have correct dietary behavior if they reported a dietary intake of at least 5 servings of fruit and vegetables per day.

A pilot study was performed on 50 randomly selected adolescents to evaluate the readability, clarity and correct sequence of the items, and reliability of the questionnaire was evaluated through the Cronbach’s alpha. Calculation of the content validity indicated the unanimous agreement with the questionnaire’s content and clarity, and test on reliability showed adequate internal consistency (Cronbach’s alpha = 0.71). No substantial corrections were made to the questionnaire; therefore, data of the pilot test were included in the final sample.

The study protocol was approved by the Ethics Committee of the Teaching Hospital of the University of Campania “Luigi Vanvitelli” (approval number 201).

### 2.4. Statistical Analysis

Statistical analyses were performed with Stata software version 15, using the svy procedures that take into account the cluster sampling design effect [23]. First, a descriptive analysis was carried out to outline the main characteristics of the sample. Then, a chi square test for categorical variables and Student’s t-test for continuous variables were conducted for bivariate analysis to assess the association between each independent variable and the different outcomes. Multivariate logistic regression models were constructed to identify factors associated with good knowledge of risk factors for cancer (no = 0; yes = 1) (Model 1), and predictors of assuming a risky behavior, namely being current smokers (no = 0; yes = 1) (Model 2), and a protective behavior, that is being physically active (no = 0; yes = 1) (Model 3). A scoring system was set for determining the level of knowledge about risk factors for cancer. A subject was classified to have a good knowledge if he/she correctly identified at least 7 out of 10 risk factors for cancers. Socio-demographics and anamnestic characteristics, as well as sources of information about cancer prevention were independent variables included in all Models. Moreover, in the Models on behaviors related to cancer (Models 2 and 3) independent variables exploring knowledge and attitudes related to cancer, as well as some cancer related behaviors were also included. A detailed description of the independent variables included in each model is reported in an additional file (Appendix A).

Odds ratios (ORs) and 95% confidence intervals (CIs) were calculated.

## 3. Results

### 3.1. Study Population

Adolescents who were invited to the survey were 900, and 871 returned the questionnaire, with a response rate of 96.8%. The average age of participants was 15.7 (range = 13–20), 45.6% and 8.8% had at least one graduate parent and one parent working in the healthcare sector, respectively, and 82.1% were under/normal weight. In particular, 18.7% of females and 29.2% of males were considered overweight/obese, respectively. Moreover, 75% of participants had personal, familiar or friend history of cancer, and 34.1% and 40.9% a second and third degree relative with cancer, respectively.

### 3.2. Knowledge and Attitudes Related to Cancer

Overall, 97% were aware of the correct definition of cancer, but only 32.5% had learnt about this definition from a physician. In total, 92.3% knew that cancer can occur at any age. Moreover, when asked to indicate the most frequent cancers in Italy, 89.3% did not include colorectal cancer, 73.7% did not include prostate cancer, 42% did not include breast cancer and 28.9% did not include lung cancer. Table 1 displays knowledge of risk factors for cancer in the study population. As expected, almost all (97.9%) reported smoking and, in descending order, alcohol drinking (69.5%), sun exposure without protection (65.7%), passive smoking (63.8%), use of tanning lamps (56.1%), consumption of high fat diets (46.3%), overweight/obesity (38.9%) and daily intake of sweets and/or sugary drinks (34%) as risk factors. Finally, only 29.5% and 20.8% knew that low intake of fruits/vegetables and poor physical activity were risk factors for cancer, respectively.

Only 19% of respondents had a good level of knowledge about the most frequent cancers in Italy and those related to smoking and alcohol use, and only 26.1% on most risk factors for cancer. Multiple logistic regression analysis showed that adolescents with both parents employed (OR = 1.84; 95% CI = 1.48–2.28) were significantly more likely to have good knowledge about risk factors for cancer, whereas those who reported to have a personal, familiar or friend history of cancer (OR = 1.49; 95% CI = 0.96–2.3, *p* = 0.066), or to have received information about cancer prevention from a physician (OR = 1.65; 95% CI = 0.97–2.83, *p* = 0.061), almost resembled a significant positive association with good knowledge about risk factors for cancer (Model 1 in Table 2).

With regard to attitudes, 95.4% of the respondents believed that cancer is a serious disease, but only 33.5% believed that it may be prevented. Moreover, one-third of respondents (33.5%) felt to be at high risk of developing cancer and only 16% were satisfied with their health status.

### 3.3. Lifestyle Behaviors Related to Cancer

When participants were asked a series of questions about their lifestyle behaviors, 41% declared that they consumed alcohol and 25.3% to be current smokers, 88.2% watched TV/used the computer for more than 2 h/day and 75.2% reported a poor physical activity. Moreover, only 19.2% consumed fruits or vegetables more than once a day, 88.1% had a daily intake of sugary drinks and 55.7% declared they exposed themselves to sun without protection. Results of the multiple logistic regression showed that older adolescents (OR = 2.01; 95% CI = 1.8–2.23), with a personal, familiar or friend history of cancer (OR = 2.49; 95% CI = 1.25–4.94), not having one parent in the healthcare sector (OR = 0.48; 95% CI = 0.37–0.62) or not being physically active (OR = 0.55; 95% CI = 0.33–0.93) were significantly more likely to be current smokers (Model 2 in Table 2), whereas male adolescents (OR = 0.33; 95% CI = 0.17–0.65), having heard the definition of cancer from a physician (OR = 1.51; 95% CI = 1.07–2.13), having one parent in the healthcare sector (OR = 1.67; 95% CI = 1.04–2.67), having a satisfactory self-reported health status (OR = 2.61; 95% CI = 1.78–3.82), not being current smokers (OR = 0.54; 95% CI = 0.33–0.9) or not having received information about cancer prevention from a physician (OR = 0.58; 95% CI = 0.4–0.84), were significantly more likely to be physically active (Model 3 in Table 2).

### 3.4. Sources of Information

Overall, 68.9% of participants had received information about cancer prevention: 43.3% from parents, followed by the internet (36.6%), healthcare professionals (24.9%) and schools (17.6%). Moreover, 83.4% stated that they felt the need for additional information about cancer prevention.

## 4. Discussion

As far as we know, the present survey is one of the few studies aiming at the evaluation of knowledge and lifestyle cancerrelated behaviors in adolescents and young adults. Most of the studies have involved adult populations [11,12,13,14] since the risk of cancer increases with age. However, it is well-known that most of the behavioral factors related to cancers may be acquired at a younger age; therefore, investigating to what extent this younger population is aware of risks and how much a risky lifestyle has been acquainted by adolescents and young adults may be of great interest to program public health interventions aimed at the reduction of the burden of cancer in older populations.

### 4.1. Knowledge and Attitudes Related to Cancer

In this survey, almost all the students were aware of the definition of cancer and that cancer can occur at any age. The findings demonstrate a higher knowledge on these aspects compared to the results of the few studies conducted in the adolescents on this topic, that showed that only 57% of the students gave the correct definition of cancer [24]; analogously, lower levels of knowledge on the age of occurrence of cancer were found in cross-sectional studies conducted in Italy [25] and in the UK [15].

Conversely, it is of concern that the most frequent cancers, except lung cancer, are not recognized by a large percentage of the participants, and even more concerning is the lack of knowledge on lifestyle-related factors, with a higher knowledge related to risk factors, such as smoking and alcohol use, and lower knowledge related to protective or healthful behaviors, such as intake of fruits/vegetables and physical activity. This pattern of knowledge on risk and protective lifestyle factors for cancer is interestingly resembled in analogous studies conducted in the adolescent population, reporting higher knowledge on smoking and alcohol [10,13,24], and lack of knowledge on the positive role of physical activity and fruit and vegetable consumption [10,13], with an even lower knowledge on role of food habits, that may be related to the higher attention to healthier eating habits and the Mediterranean diet in Italy.

This is of note, since knowledge is one of the predisposing factors that may provide the motivation to adopt healthful or risky behaviors or lifestyles and that can often be influenced by educational interventions. The finding that risky behaviors were more frequently recognized by students as related to cancer compared to healthful behaviors may be due to the more consolidated knowledge on smoking and alcohol as risk factors for cancer, compared to the knowledge on the role of diet and physical activity, but it could also be a consequence of the low awareness on the preventability of cancer shown by a large proportion of the surveyed students, ignoring that subjects may have an active role in the prevention of cancer by adopting healthful behaviors with regard to dietary habits and physical activity. Health promotion interventions aimed at increasing knowledge should primarily focus on lifestyle protective behaviors, that appear to be less known in this population, as well as on successes that have been obtained in cancer prevention activities.

Our findings showed that knowledge was higher in those who reported the history of cancer in an influential person, and confirm the positive role that healthcare professionals may have on providing correct health-related information, since those who reported to have been informed on cancer by a physician had a higher knowledge. Indeed, previous studies conducted by some of us have reported significant associations between knowledge on several health-related topics and information received by physicians [26,27,28].

### 4.2. Lifestyle Behaviors Related to Cancer

Although most of the participants consider cancer a serious disease, and one third are considered to be at high risk of developing cancer, the results of our study confirmed a wide diffusion of an unhealthy lifestyle among adolescents, that once consolidated during adolescence, may have substantial consequences in adulthood, including the risk of developing cancer. The consistently widespread smoking habit and alcohol consumption is worrying, although it is similar in respect to analogous surveys conducted in the adolescent populations [29] or those of higher ages [30,31,32,33]. Equally, or even more, concerning is the result on the poor physical activity and insufficient consumption of fruits or vegetables, although analogous to other reported frequencies in similar studies [31,33,34]. Indeed, these data have been confirmed by a pooled analysis of surveys conducted in a large number of countries, reporting that more than 80% of teenagers aged 11 to 17 do not reach the levels of physical activity recommended for a healthy lifestyle [35]. These findings suggest that future intervention in this population should target multiple lifestyle behaviors simultaneously to reduce the burden of cancer and other of lifestyle-related diseases in adulthood.

The choice to investigate one health-risk behavior (being current smoker) and one health-promoting behavior (physical activity) and their predictors through multivariate analysis was driven by the theory that behaviors may be clustered into health-risk behaviors (such as alcohol and nicotine consumption) and health-promoting behaviors (such as physical activity and nutrition) [36], and that interventions targeting one behavior have been reported to facilitate changes in other clustered behaviors [37]. This has implications for the design of educational interventions, since a strategy targeting one behavior may have effects on multiple health behaviors within the same cluster [38]. The findings of the multivariate analysis showed that having one of the parents working in the healthcare sector was positively associated with not being a current smoker, suggesting that having a privileged access to competent health-related information may have a positive role to reduce unhealthy habits in adolescents. This is expected since health knowledge favors the maintenance or improvement of health-related behaviors and quality of life by making the appropriate judgments on disease prevention and health promotion Moreover, the finding that current smokers perform poor physical activity confirms that unhealthy and healthy behaviors tend to cluster in different subgroups of adolescents, and demand approaches targeted to the specific domain of health-risk or health-promoting behaviors. The profile of the adolescent performing adequate physical activity showed a consistent pattern with previous results confirming that the information on cancer provided by physicians promotes the engagement in healthy behaviors. Moreover, the finding that these subjects are more likely to be satisfied by their health status is in line with previous studies reporting that high physical activity has a positive impact on adolescent perceived health, risk behaviors and mental health, and that increased levels of physical activity can play a vital role in prevention of health risks and in adolescent health promotion [39]. The importance of an active role on the choice of a healthy lifestyle behavior has also been reported for dietary habits demonstrating that physical restriction of food in children was associated with higher nutrition risk, whereas positive comments were associated with a lower nutrition risk in children [40], suggesting that positive encouragement is more effective compared to restriction.

### 4.3. Sources of Information

It is worth noting that although more than two thirds of the participants were reported to have received information on cancer prevention, the vast majority perceived the need for further information, suggesting that the interest on this topic may encourage the implementation of interventions aimed at the promotion of cancer prevention knowledge and of the appropriate skills to engage in multiple healthy behaviors. It is interesting to acknowledge that almost half of the sample declared that they received information from the parents, whilst only a quarter received information from physicians; parents have been reported to be a more frequent source of information on other health-related topics for adolescents [27]. This finding underlines that there is place for a greater commitment for health professional in the improvement of adolescents’ knowledge and self-confidence on cancer prevention. 

### 4.4. Limitations

Although these findings are helpful to have insight and stimulate research on knowledge, attitudes and lifestyle behaviors related to cancer prevention in young populations, the results should be interpreted considering several potential limitations. First, it should be noted that the study was cross-sectional; therefore, there was no prospective evaluation of the effect of predictors on knowledge and health related behaviors, thus limiting the assessment of cause–effect relationships. Second, the survey was carried out only in one geographic area of Italy; therefore, generalization of results should be made with caution. Third, we dichotomized the outcomes of interest, and therefore there may have been some loss of information. However, this choice allowed us to describe behaviors according to WHO recommendations [20] and to have a more simple way to describe “good knowledge”. Finally, since it explored reported lifestyle behaviors, the answers might be different from actual behavior with the risk of the underestimation of sensible behaviors, such as alcohol consumption, with the potential for social desirability bias. However, since confidentiality of the collected data had been assured to participants, we believe that this issue might have been somewhat mitigated. Despite these limitations, the research involved a well-designed survey and had a high response rate, assuring to a great extent the representativeness of the population. 

## 5. Conclusions

Our study showed that adolescents had a low level of knowledge about most risk factors of cancer and lifestyle cancer-related behaviors. It is widely agreed that many causes of cancer are modifiable, and that changes in behavior could prevent them. Therefore, these findings support the need for implementing health educational intervention among younger about healthy behaviors and the prevention of cancer, with the aim of improving their level of knowledge and promoting behavior change.

## Figures and Tables

**Table 1 ijerph-17-08294-t001:** Knowledge of cancer risk factors in agreement with the European Code against Cancer [17].

Risk Factors for Cancer	Total	
	*N* = 871	
	*N*	%
Smoking	853	97.9
Alcohol drinking	605	69.5
Sun exposure without protection	572	65.7
Passive smoking	556	63.8
Use of tanning lamps	489	56.1
High fat diets	403	46.3
Overweight/obesity	339	38.9
Daily intake of sweets and/or sugary drinks	296	34
Low intake of fruits/vegetables	257	29.5
Poor physical activity	181	20.8

**Table 2 ijerph-17-08294-t002:** Logistic regression models results.

Variable	OR *	Linearized SE ^+^	95% CI ^σ^	*p*-Value
**Model 1.** Good knowledge about risk factors for cancer (sample size = 852)				
Having both parents employed				
No	1 **			
Yes	1.84	0.16	1.48–2.28	<0.001
Having personal, familiar, or friend history of cancer				
No	1 **			
Yes	1.49	0.26	0.96–2.3	0.066
Information about cancer prevention received from a physician				
No	1 **			
Yes	1.65	0.36	0.97–2.83	0.061
Age in years, ordinal (13–15 = 1; 16–17 = 2; 18–20 = 3)	1.27	0.2	0.87–1.87	0.176
Need of additional information about cancer prevention				
No	1 **			
Yes	1.01	0.001	0.99–1.01	0.739
Gender				
Male	1 **			
Female	0.99	0.07	0.83–1.19	0.913
**Model 2.** Being current smokers (sample size = 852)				
Age in years, ordinal (13–15 = 1; 16–17 = 2; 18–20 = 3)	2.01	0.09	1.8–2.23	<0.001
Having personal, familiar, or friend history of cancer				
No	1 **			
Yes	2.49	0.7	1.25–4.94	0.018
Having one parent in the healthcare sector				
No	1 **			
Yes	0.48	0.05	0.37–0.62	<0.001
Knowledge about most frequent cancers in Italy and those related to smoking and alcohol use				
No	1 **			
Yes	0.59	0.2	0.26–1.34	0.167
Being physically active				
No	1 **			
Yes	0.55	0.12	0.33–0.93	0.031
Knowing that insufficient intake of fruits/vegetables, daily intake of sweets and/or sugary drinks, and high fat diets were risk factors for cancer				
No	1 **			
Yes	0.54	0.2	0.22–1.32	0.145
Consuming alcohol				
No	1 **			
Yes	0.74	0.14	0.47–1.17	0.158
Having a daily intake of sugary drinks				
No	1 **			
Yes	1.22	0.17	0.86–1.74	0.214
Perception to be at risk of developing cancer				
Low	1 **			
High	1.31	0.34	0.69–2.48	0.337
Having a satisfactory self-reported health status				
No	1 **			
Yes	0.82	0.17	0.5–1.36	0.383
Having at least a graduate parent				
No	1 **			
Yes	0.9	0.13	0.64–1.29	0.515
Having heard the definition of cancer from a physician				
No	1 **			
Yes	0.92	0.16	0.61–1.4	0.657
Knowledge that poor physical activity and overweight/obesity are risk factor for cancer				
No	1 **			
Yes	1.06	0.24	0.61–1.84	0.794
Knowledge that smoking is a risk factor for cancer				
No	1 **			
Yes	0.84	0.66	0.12–5.69	0.830
Information about cancer prevention received from a physician				
No	1 **			
Yes	0.95	0.29	0.45–2.01	0.886
Gender				
Male	1 **			
Female	0.99	0.11	0.75–1.32	0.959
Consuming at least 5 servings of fruits and vegetables per day				
No	1 **			
Yes	0.99	0.14	0.71–1.39	0.969
**Model 3.** Being physically active (sample size = 852)				
Gender				
Male	1 **			
Female	0.33	0.09	0.17–0.65	0.007
Age in years, ordinal (13–15 = 1; 16–17 = 2; 18–20 = 3)	0.68	0.12	0.44–1.05	0.074
Having heard the definition of cancer from a physician				
No	1 **			
Yes	1.51	0.21	1.07–2.13	0.027
Having a satisfactory self-reported health status				
No	1 **			
Yes	2.61	0.4	1.78–3.82	0.001
Being current smokers				
No	1 **			
Yes	0.54	0.11	0.33–0.9	0.026
Having one parent in the healthcare sector				
No	1 **			
Yes	1.67	0.32	1.04–2.67	0.038
Having received information about cancer prevention from a physician				
No	1 **			
Yes	0.58	0.09	0.4–0.84	0.011
Having at least a graduate parent				
No	1 **			
Yes	1.25	0.13	0.97–1.6	0.073
Knowledge about most frequent cancers in Italy and those related to smoking and alcohol use				
No	1 **			
Yes	1.18	0.09	0.97–1.43	0.079
Perception to be at risk of developing cancer				
Low	1 **			
High	1.17	0.22	0.73–1.85	0.446
Knowledge that smoking is a risk factor for cancer				
No	1 **			
Yes	0.71	0.34	0.22–2.27	0.501
Consuming alcohol				
No	1 **			
Yes	0.91	0.12	0.65–1.27	0.523
Consuming at least 5 servings of fruits and vegetables per day				
No	1 **			
Yes	1.15	0.25	0.66–1.98	0.563
Knowledge that poor physical activity and overweight/obesity are risk factor for cancer				
No	1 **			
Yes	1.07	0.18	0.7–1.64	0.713
Having personal, familiar, or friend history of cancer				
No	1 **			
Yes	1.03	0.21	0.62–1.69	0.895
Knowing that insufficient intake of fruits/vegetables, daily intake of sweets and/or sugary drinks, and high fat diets were risk factors for cancer				
No	1 **			
Yes	1.02	0.42	0.37–2.77	0.966
Having a daily intake of sugary drinks				
No	1 **			
Yes	1.01	0.26	0.53–1.88	0.984

* Odds Ratio. ^+^ Standard Error. **^σ^** Confidence Interval. ** Reference category.

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
