# Peer review of "Cancer Prevention: Knowledge, Attitudes and Lifestyle Cancer-Related Behaviors among Adolescents in Italy"

_ijerph, 2020, doi:10.3390/ijerph17228294_

Round 1
Reviewer 1 Report
The need for the study and its potential contribution is well justified, its design is appropriate, and it provides enough information on rigorous sampling, data processing as well as compliance with ethical standards.
However, I suggest improving the explanation of validity assessment of the questionnaire. In the Materials and methods section the authors indicate that a pilot study was performed to evaluate the readability and reliability of the questionnaire, but they do not explain the procedure and statistical analysis used (for example, if an incomprehensible item was modified or if Crombach alpha was used to assess reliability). Also, I think that is not recommended to include data of the pilot study in the final sample. Moreover, I suggest providing all items of the questionnaire in supplementary file.
In other hand, I suggest to review the statistical analysis section. The wording is cumbersome, and perhaps it would be better if it was structured in subsections for each model.
Author Response
Reviewer 1
I suggest improving the explanation of validity assessment of the questionnaire. In the Materials and methods section the authors indicate that a pilot study was performed to evaluate the readability and reliability of the questionnaire, but they do not explain the procedure and statistical analysis used (for example, if an incomprehensible item was modified or if Crombach alpha was used to assess reliability).
In response to this point, during the pilot study no substantial modifications of the questionnaire were needed, and we have now reported this result in the text. Moreover, we used the Cronbach alpha to assess reliability and the results of the test have now been included in the text (Lines 114-118).
Also, I think that is not recommended to include data of the pilot study in the final sample.
In response to this point, we agree with the reviewer that in many circumstances it is not appropriate to include the results of the pilot study in the main analysis. However, in this case no substantial corrections were made to the questionnaire, and we believe the inclusion of the results of the pilot study would only increase the amount of the information gathered. We have now specified in the text the reason why we did not exclude the results of the pilot study from the main analysis (Lines 118-119).
Moreover, I suggest providing all items of the questionnaire in supplementary file.
As suggested, we have included the questionnaire used in a Supplementary file, and this has been pointed out in the methods section (Line 97).
In other hand, I suggest to review the statistical analysis section. The wording is cumbersome, and perhaps it would be better if it was structured in subsections for each model.
As suggested, we have revised the statistical analysis section (Lines 132-137). Moreover, we have structured subsections for each model and reported them in an Appendix (Line 154).

Reviewer 2 Report
General Comments
The authors aim at exploring knowledge, attitudes, and lifestyle behaviors related to cancer in a sample of adolescents aged 13-20 years.
Three binary endpoints are considered in the multivariate analyses: 1) Knowledge of risk factors for cancer (no/yes), 2) current smoker (no/yes),
and physically active (no/yes). Generally, the topic of cancer prevention is interesting and relevant. However, there are a number of conceptual
and methodological issues, which need to be resolved. More specifically, the authors should address the following items.
Specific comments
Page 2, line 51-57: Two-stage cluster sampling was used to select 13-20 years old adolescents attending public secondary schools.
Please indicate if and how the design effect resulting from the sampling procedure was considered in the analyses.
Page 2, line 66 and page 3, line 127: BMI
Reporting the overall mean BMI for a sample of adolescents between 13-20 is inadequate (page 3, line 127). BMI is depending on gender and age and a pediatric BMI calculation and classification system should be used for this population. Please revise.
Page 2, lines 82-85 and page 3, lines 98-103: In the questionnaire section, the authors indicate that physical activity and sedentary behavior have been assessed using time scales. However, in the multivariate statistical analyses, a binary endpoint for physical activity has been used. The authors should clarify a) what instruments have been used to assess physical activity (e.g. IPAQ), b) what cutoffs have been used to classify the population, and c) provide evidence on why such a classification is applicable and valid over a population of adolescents with a very diverse age range and hence substantial differences in body development. Moreover, the authors should d) discuss potential effects of information loss due to the scale reduction.
Similarly, the authors should clarify points a) – d) for sedentary and dietary behavior. For the level of knowledge about risk factors (page 3, lines 101-103) the authors should at least cover d) and inform the reader why it was necessary to collapse the scale to a binary indicator and potentially induce information loss.
Since sedentary behavior and physical activity are used in the same models (as binary indicators), the authors also need to provide information on what exactly constitutes the difference between these two variables and whether these indicators measure different aspects. Furthermore, the authors should explain how someone can express “no sedentary behavior” or use a more appropriate category name.
Page 3, line 103-104: The authors claim that three age categories have been used: “11-15; 16-17; 18-20. Since your sample included a population of 13-20 year olds, the first category should read “13-15”. Please correct.
Moreover, in the multivariate analyses (Table 2, page 4) the reference to age groups is ambiguous. The authors refer to “younger adolescents” and “older adolescents” – without any knowledge on the reference group, this information cannot be interpreted. Moreover, it is good methodological and statistical practice to continue referring to category names and classification that were previously given. Please refer to your three age categories and tell the reader which one is the reference category.
Page 4, Table 2: This table is weird, at best confusing. Firstly, using “not” in variable names is not always the best choice, i.e. having variables names such as “physically active”, “consuming alcohol” or “sedentary behavior” should be preferred over “not physically active”, “not consuming alcohol” or “no sedentary behavior”. More importantly, that is what the authors did in the first place! E.g. they defined being physical active (0=no, 1=yes) on page 3 lines 102-103/117. Having this in mind, the reader would expect to find EXACTLY this variable name in Model 2 (the same applies to other variable names that have been changed). However, we are confronted with “Not being physically active” and it is not clear at that moment to what the latter variable is referring because it has not been defined! Please prevent this confusing name
changes in all MODELS (Table 2) and refer to the variable names that have been previously defined.
More importantly, it is currently impossible to fully assess whether the interpretation of the parameter estimates is correct and consistent with expectations. Consider Model 2: according to page 3, lines 114-118, physical activity, sedentary behavior, and alcohol consumption have been coded (0=no, 1=yes). The corresponding reported OR in Model 2 are: 0.43, 0.75, and 0.65. Using your initial coding scheme for the variables, this would indicate that 1) being physically active is associated with a lower odds for smoking; 2) alcohol consumption is associated with a lower odds for smoking, and 3) sedentary behavior is associated with a lower odds of smoking.
However, this not entirely what has been reported on page 5, lines 159-170 where these rather surprising results for alcohol and sedentary behavior do not appear. Please clarify and discuss.
Last but not least, the manuscript is primarily concerned with RISK factors. It may therefore be worthwhile considering to recode some of your variables so that they consistently reflect risks. E.g. instead of using a variables such as “physical activity” use the variable “physical inactivity”, instead of using “knowledge about X” use “lack of knowledge about X” and code it accordingly. This approach will prevent you from using hard to grasp constructs involving “NOT”.
Author Response
Reviewer 2
Specific comments
Page 2, line 51-57: Two-stage cluster sampling was used to select 13-20 years old adolescents attending public secondary schools.
Please indicate if and how the design effect resulting from the sampling procedure was considered in the analyses.
In response to this point, we have indeed taken into account the design effect due to the two-stage cluster sampling, and we have now specified it in the sample size definition paragraph (Lines 60-63).
Page 2, line 66 and page 3, line 127: BMI
Reporting the overall mean BMI for a sample of adolescents between 13-20 is inadequate (page 3, line 127). BMI is depending on gender and age and a pediatric BMI calculation and classification system should be used for this population. Please revise.
As suggested, the results on BMI have now been modified and reported considering the BMI percentile distribution of the Southern Italian population for the same age and sex groups (Lines 163-164). Moreover, how BMI was evaluated has been reported in the Methods section (Lines 98-101).
Page 2, lines 82-85 and page 3, lines 98-103: In the questionnaire section, the authors indicate that physical activity and sedentary behavior have been assessed using time scales. However, in the multivariate statistical analyses, a binary endpoint for physical activity has been used. The authors should clarify a) what instruments have been used to assess physical activity (e.g. IPAQ), b) what cutoffs have been used to classify the population, and c) provide evidence on why such a classification is applicable and valid over a population of adolescents with a very diverse age range and hence substantial differences in body development. Moreover, the authors should d) discuss potential effects of information loss due to the scale reduction.
As suggested, we have specified the instrument we used to assess physical activity, and the relative cut-off. Moreover, these measures have been developed for all the considered age groups, and are reported as dichotomous variables (Lines 102-112). In addition, since our aim was to identify predictors of healthy or risky behaviors, a dichotomous variable would be more suitable for this objective, although some loss of information may occur (Lines 108-109). We have now explained this choice in the limitation section (Lines 320-323).
Similarly, the authors should clarify points a) – d) for sedentary and dietary behavior.
As suggested, we have specified the instrument we used to assess sedentary and dietary behaviors, and the relative cut-offs (Lines 102-112). Also in these cases, we used definitions developed for the adolescents, that are reported as dichotomous variables (Lines 108-112).
For the level of knowledge about risk factors (page 3, lines 101-103) the authors should at least cover d) and inform the reader why it was necessary to collapse the scale to a binary indicator and potentially induce information loss.
In response to this point, the choice to collapse knowledge into two categories has been driven by the same reasons reported for the other outcome variables, and this has been explained in the limitations section (Lines 320-323).
Since sedentary behavior and physical activity are used in the same models (as binary indicators), the authors also need to provide information on what exactly constitutes the difference between these two variables and whether these indicators measure different aspects. Furthermore, the authors should explain how someone can express “no sedentary behavior” or use a more appropriate category name.
In response to this point, we agree with the referee that these two variables tend to measure the same behavior and have decided to construct the model excluding sedentary behavior, with only physical activity as an independent variable.
Page 3, line 103-104: The authors claim that three age categories have been used: “11-15; 16-17; 18-20. Since your sample included a population of 13-20 year olds, the first category should read “13-15”. Please correct.
As suggested, we have corrected the typo.
Moreover, in the multivariate analyses (Table 2, page 4) the reference to age groups is ambiguous. The authors refer to “younger adolescents” and “older adolescents” – without any knowledge on the reference group, this information cannot be interpreted. Moreover, it is good methodological and statistical practice to continue referring to category names and classification that were previously given. Please refer to your three age categories and tell the reader which one is the reference category.
As suggested, we have completely rewritten the results of the model in Table 2 including the names of the variables and their respective categories. As regards to age, we have included the variable as ordinal and therefore the odds ratio indicates the increase or decrease of the risk for each category. We have specified that the variable was ordinal both in the Table 2 and in the supplemental file describing model building (S2).
Page 4, Table 2: This table is weird, at best confusing. Firstly, using “not” in variable names is not always the best choice, i.e. having variables names such as “physically active”, “consuming alcohol” or “sedentary behavior” should be preferred over “not physically active”, “not consuming alcohol” or “no sedentary behavior”. More importantly, that is what the authors did in the first place! E.g. they defined being physical active (0=no, 1=yes) on page 3 lines 102-103/117. Having this in mind, the reader would expect to find EXACTLY this variable name in Model 2 (the same applies to other variable names that have been changed). However, we are confronted with “Not being physically active” and it is not clear at that moment to what the latter variable is referring because it has not been defined! Please prevent this confusing name
changes in all MODELS (Table 2) and refer to the variable names that have been previously defined.
More importantly, it is currently impossible to fully assess whether the interpretation of the parameter estimates is correct and consistent with expectations. Consider Model 2: according to page 3, lines 114-118, physical activity, sedentary behavior, and alcohol consumption have been coded (0=no, 1=yes). The corresponding reported OR in Model 2 are: 0.43, 0.75, and 0.65. Using your initial coding scheme for the variables, this would indicate that 1) being physically active is associated with a lower odds for smoking; 2) alcohol consumption is associated with a lower odds for smoking, and 3) sedentary behavior is associated with a lower odds of smoking.
However, this not entirely what has been reported on page 5, lines 159-170 where these rather surprising results for alcohol and sedentary behavior do not appear. Please clarify and discuss.
In response to this point, we agree with the referee that the way we presented the results of the models may be confusing, and we have now presented the data including all the categories of the variables (Table 2). Moreover, we agree that the results on alcohol and sedentary behavior appear surprising, but indeed they were not statistically significant, so, as expected, no association was found.
Last but not least, the manuscript is primarily concerned with RISK factors. It may therefore be worthwhile considering to recode some of your variables so that they consistently reflect risks. E.g. instead of using a variables such as “physical activity” use the variable “physical inactivity”, instead of using “knowledge about X” use “lack of knowledge about X” and code it accordingly. This approach will prevent you from using hard to grasp constructs involving “NOT”.
In response to this point, we agree with the referee that we have used the word risk factors, but indeed we were interested in behaviors that may influence the risk of cancer, both increasing and decreasing the risk, and this is the reason why we modeled both smoking and physical activity, that is a behavior that is associated to an increased risk of cancer, and another that is related to a decrease in the risk of cancer. This has also been discussed in the discussion section. Moreover, to avoid misunderstanding we have now rephrased throughout the text and in the title the word risk factor with “cancer-related behavior”. There fore, we have changed the title “Cancer prevention: knowledge, attitudes and risky lifestyle among adolescents in Italy” in “Cancer prevention: knowledge, attitudes and lifestyle cancer related behaviors among adolescents in Italy”.

Round 2
Reviewer 2 Report
I would like to thank the authors for consindering my suggestions. The manuscript has been substantially improved. There remains a minor point that should be addressed by the autors. In my first review I asked for a clarification of if and how the analyses have taken into account the design effect resulting from cluster sampling. In their response and the revised manuscript (Lines 60-63, sample size definition paragraph), the authors pointed out that they took into account the design effect when calculating the sample size. While this is indeed very important, it is still not clear whether they used design specific calculations of variances in the ANALYSES that accounted for the complex design of their survey. Thank you in advance for clarifying this point.
Author Response
In response to this point, in the first version of the manuscript, we took into account the cluster sampling design effect only in the definition of the sample size. According to your suggestion, we have now repeated the analysis, using the STATA procedure that allows to correct variances according to the cluster sampling design. This has been explained in the Methods section (), and related modifications in the results (Lines 159-165; 184-196), discussion (Lines 238-239; 272-274), Table 2, and abstract (Lines 22-24) have been reported in the manuscript.
